# Eating Occasions, Obesity and Related Behaviors in Working Adults: Does it Matter When You Snack?

**DOI:** 10.3390/nu11102320

**Published:** 2019-10-01

**Authors:** Wendy E. Barrington, Shirley A. A. Beresford

**Affiliations:** 1Department of Child, Family, and Population Health Nursing, University of Washington, Seattle, WA 98195, USA; 2Department of Epidemiology, University of Washington, Seattle, WA 98195, USA; sberesfrd@uw.edu; 3Cancer Prevention Program, Fred Hutchinson Cancer Research Center, Seattle, WA 98195, USA

**Keywords:** snacking, obesity, obesogenic, behaviors

## Abstract

Reported relationships between frequency, type, and timing of eating occasions and obesity-risk among adults are mixed while associations with obesogenic eating behaviors remain unexplored. The Physical Activity and Changes in Eating (PACE) study was a group-randomized controlled trial to prevent weight gain among 34 small worksites in Seattle from 2005–2009. Baseline surveys assessed body mass index (BMI), obesogenic eating behaviors (e.g., fast food and distracted-eating), and eating occasions (i.e., snacks and meals) among 2265 employees. BMI and waist circumference were measured on a subset (*n* = 567). Time-periods for analyses included: morning (12:00 a.m. to 10:59 a.m.), mid-day (11:00 a.m. to 4:29 p.m.), and evening (4:30 p.m. to 11:59 p.m.). Multilevel linear models estimated associations between snack timing, obesity, and related behaviors while adjusting for meal timing, gender, and worksite random effects. Greater morning snacking was associated with increased fruit and vegetable consumption, while greater evening snacking was associated with higher BMI, higher obesogenic dietary index (intake of fast food, French fries, and soft drinks), and higher percent time eating while distracted. Associations with mid-day snacking were mixed. Patterns of association were consistent across repeated and objective measures. Findings suggest that evening snacking is more detrimental to healthy weight compared to snacking at other times of day. Reducing evening snacks may be an important and simple message for population-level obesity prevention efforts.

## 1. Introduction

Obesity continues to threaten the health of U.S. adults—almost 40% are affected and obesity prevalence among women has continued to increase [1]. Alarmingly, obesity prevention efforts, including dietary modification, have largely been unsuccessful in maintaining long-term weight reduction [2]. Our understanding of what dietary interventions are most effective is limited not only by how diet has been measured, but also by what messages have been promoted for dietary change. Most studies have focused on communicating relationships between nutrient intake and obesity-risk. Focus has shifted recently towards understanding the composition of ‘obesogenic’ eating patterns and dietary behaviors. This approach is not only more easily communicated to the public in the form of dietary guidelines, but also accounts for how nutrients are consumed together as foods and meals [3].

Eating occasions is a term used to describe the ingestion of any type of meal or drink which can be further categorized by amount and type of food consumed as well as the time of day it was consumed [3]. Some [4,5], but not all, evidence supports associations between greater frequency of eating occasions and obesity [4,5]. However, our understanding of the relationships between type and timing (i.e., pattern) of eating occasions and obesity is less clear due to differences in how eating occasions are defined across studies [3,4]. Some studies pre-define the eating occasion (e.g., breakfast, morning snack, lunch, afternoon snack, dinner, evening snack, overnight) [3,4] for the respondent. Such definitions assume intake occurs at implied culturally-relevant times and usually provide examples of relevant foods for each type of eating occasion. Other studies may rely on other factors associated with the eating occasion (e.g., duration, timing, portion size, energy density, % contribution to daily energy intake) for the researcher to define the type of eating occasion [3,4]. For snacks in particular, some U.S. studies define these occasions by intake of specific foods that tend to be energy-dense [3,4]. Despite these methodological differences, several reviews consistently support associations between greater snacking frequency [4,5,6,7,8], as well as eating later in the day [6,9], with higher obesity risk. However, most studies have not evaluated the specific contribution of snacking while accounting for other eating occasions [3]. Such analyses could more closely model associations between what may be considered ‘extra’ eating occasions and obesity. Furthermore, understanding the relationship between snacking pattern and obesogenic behaviors is also warranted [4]. The effects of snacking may depend not just on the amount of food consumed, but behaviors that determine ‘what’ and ‘how’ food is consumed [10,11,12].

The Promoting Activity and Changes in Eating (PACE) study provides a unique opportunity to address these methodologic issues using a novel grid assessment method [13,14] for measuring eating occasions and patterns. This method allows the respondent to categorize their own eating occasion as well as document when intakes occurred. The objective of this study, specifically, is to evaluate snacking patterns in relation to obesity (i.e., body mass index, waist circumference), and to obesity-related behaviors (i.e., fruit and vegetable consumption, obesogenic dietary index score, and distracted-eating). We hypothesize that that higher snacking, especially later in the day, will be associated with obesity among this sample of adults. Further, if their snacking patterns are associated with other obesogenic dietary behaviors like fast food or soda intake, this may support directing behavior change goals to reduce snacking, as a simple and more easily accepted strategy, in future obesity prevention interventions.

## 2. Materials and Methods 

### 2.1. Study Population

The Promoting Activity and Changes in Eating (PACE) study was a large group-randomized weight-gain prevention trial among approximately 3000 individuals in 34 worksites in the Seattle Metropolitan area. Eligible worksites that employed between 40 and 350 workers were identified using U.S. Standard Identification Classification two-digit codes; worksite categories included: manufacturing, transportation or utilities, personal services, household and miscellaneous services, and non-classifiable establishments. Eligibility criteria included having: 1) a high proportion of sedentary employees; 2) a low turn-over rate during the previous two years; and 3) a low proportion of non-English speaking employees. Detailed description of the PACE study has been reported elsewhere [15].

All employees of participating worksites with fewer than 150 employees and a random subsample of 125 employees within worksites with greater than 150 employees were asked to complete a standard questionnaire assessing self-reported dietary and physical activity behaviors, height, weight, and demographic information at baseline (2005–2007). An independent sample of employees, derived in the same way, were invited to complete the follow-up questionnaire (2007–2009). At baseline, 3054 employees within 34 worksites provided survey data; a random ‘intensive assessment’ subsample of 622 employees was also invited to provide additional physical measures including measured height, weight, and waist circumference. The present analysis uses survey data from 2389 individuals without missing data from the eating occasion grid or gender at baseline. Of these individuals, 1151 provided survey data at 2-year follow-up as well as physical measurements as part of intensive assessment subsamples: 568 at baseline and 268 at follow-up. This study was approved by the Fred Hutchinson Cancer Research Center Institutional Review Board.

### 2.2. Eating Occasions and Patterns

The time and type of eating occasion were collected using a pattern matrix or grid adapted from the work of Berteus Forslund and colleagues [13,14]. In brief, respondents wrote in the time of every eating occasion and checked which type of meal that represented. The choices were: main meal (e.g., cooked dish, hearty soup with bread, Chef salad with bread, pizza); light meal/breakfast (e.g., cooked or cold cereals, simple soup, side salad); snack meal (e.g., cookie, slice of cake/pie, energy bar, fruit, candy, ice cream); or drink only (e.g., coffee, tea, soft drink, juice, milk, beer, wine). Respondents were not limited in the number of entries they could submit on the grid. Our adaptation of the grid is included as Appendix A.

#### 2.2.1. Snack Frequency

The distribution of snack times over the day was visually examined to identify natural cut-points. Based on this approach, three major snack time periods were identified including: morning (12:00 a.m. to 10:59 a.m.), mid-day (11:00 a.m. to 4:29 p.m.), and evening (4:30 p.m. to 11:59 p.m.). The number of snack occasions was summed within each snack period to obtain the frequency and grouped into frequency categories (i.e., ‘0’, ‘1’, ‘2+’).

#### 2.2.2. Intake of Main Meal

Presence or absence of a main or light meal within each of the distinct snack periods referenced above resulted in three binary variables: morning meal, mid-day meal, and evening meal.

### 2.3. Outcome Variables

#### 2.3.1. Body Mass Index (BMI)

At both baseline and follow-up, height, and weight were assessed via self-report on the survey, and were measured in the intensive assessment subsample by trained study personnel using a stadiometer and scale, respectively. BMI was calculated as weight (kg) divided by height (m^2^) using both self-reported and physically measured data in two separate variables for these analyses.

#### 2.3.2. Waist Circumference

Waist circumference was assessed via physical measurement during the same intensive assessment within worksites by study personnel at baseline and follow-up. Values are reported in cm.

#### 2.3.3. Fruit and Vegetable Servings Per Day

Increased daily servings of fruits and vegetables is a behavior promoted heavily by public health professionals as it has been found to be inversely associated with obesity [16]. Components of fruit and vegetable consumption were assessed using the National Cancer Institute’s seven-item 5-a-Day fruit and vegetable assessment tool [17]. Total servings of fruit and vegetables was calculated by summing all components except frequency of eating fried potatoes or French fries. 

#### 2.3.4. Index of Obesogenic Dietary Behaviors Per Week

Using reduced rank regression, several dietary behaviors were identified that best predicted obesity among PACE participants [18]. The index is a simple average of weekly frequency of three intakes: french fries, soft drinks, and fast food meals. Frequency of eating fried potatoes or french fries was assessed by one component of the NCI 5-a-Day fruit and vegetable assessment tool [17]. Frequency of fast-food meals was assessed by the item, “Thinking about how often you eat out, how many times in a week or month do you eat breakfast, lunch, or dinner in a place such as McDonald’s^®^, Burger King^®^, Wendy’s^®^, Arby’s^®^, Pizza Hut^®^, or Kentucky Fried Chicken^®^?”) [19,20]. Responses were given as times per week or times per month. All responses were converted to times per week. Average weekly soft-drink intake was assessed via the item, “How often do you drink soft drinks or soda pop (regular or diet)?” [21]. Response options were: “Never”, “Less than once a week”, “About once a week”, “2-5 times per week”, “About once a day”, and “2 or more times per day.”

#### 2.3.5. Task-Eating

This construct connotes a level of distraction, or lack of eating awareness or mindfulness, which has also been linked to obesity [22,23]. Task-eating was assessed via single-item, “How often do you eat food (meals or snacks) while doing another activity, for example, watching TV, working at a computer, reading, driving, or playing video games?” [10]. Response options were on a five-point Likert scale ranging from 1 = ‘Never’ to 5 = ‘Always’. 

### 2.4. Covariates

Individual-level factors in models were: age, gender, race/ethnicity (where ‘Other’ consolidates smaller racial/ethnic subgroups including: Native Alaskan/American Indian and Pacific Islander/Native Hawaiian groups), and education. To further adjust for lifestyle differences, models also included adjustment for manual (i.e., machine operators, mechanics/technicians, service workers, tradesmen, or laborer) or non-manual occupation and for leisure-time physical activity of at least 10 minutes in duration via the Godin–Shephard Leisure-Time Physical Activity Questionnaire [24]. The questionnaire estimates the frequency of exercise bouts (i.e., vigorous, moderate, and light), multiplies each bout by the corresponding metabolic equivalents (i.e., 9, 5, and 3), and then sums these components to create an intensity-weighted score (i.e., leisure score index) that corresponds to a metabolic equivalent of task frequency per week [24].

### 2.5. Statistical Analysis

To examine associations between snacking and obesity-related outcomes, frequency categories of snack occasions was regressed against outcome variables adjusted for covariates and presence/absence of a main or light meal for the corresponding snack period. Continuous outcome variables were log-transformed to account for skewness during analyses and marginal means were calculated for frequency categories within snack periods. All analyses were conducted using Stata SE version 13.0 (StataCorp, College Station, TX, USA).

## 3. Results

Baseline demographic and eating occasion data among PACE participants are summarized in Table 1. The mean age of participants was 43 years and participants were primarily white, had greater than a high school education, worked in white- or pink-collar jobs (i.e., non-manual occupation), and were sufficiently active according to an established cut-point (≥ 24 leisure score index) for the Godin Shepard Leisure-Time Physical Activity Questionnaire [25]. Participants reported eating about 2 main meals, 1 light meal, 1.5 snacks, and 1 solo drink per day. About 80 percent of respondents reported snacking at least once. In the morning, 66.6% reported eating a meal whereas meal intake was higher during later time-periods (90.5% and 93.9%, respectively).

Correlations between eating occasions, dietary behaviors, and obesity outcomes at baseline are presented in Table 2. There was virtually no correlation between snacking variables nor meal variables. However, there was a modest negative correlation between morning snacking and morning meals and between mid-day snacking and mid-day meals while a modest correlation between morning meals and mid-day snacking was evident. Among dietary behaviors, there was modest negative correlation of fruit and vegetable intake with the obesogenic dietary index as well as with distracted eating. Conversely, both self-reported and measured obesity variables were highly correlated.

Outcome measures at baseline and follow-up are presented in Table 3. On average, participants were overweight/obese, did not meet the recommended number of five servings of fruit and vegetable per day, ate french fries, fast food, and soft-drinks about three times per week, and had high-levels of distracted eating (Table 3). Repeated measures at baseline and follow-up were highly correlated for all variables except distracted-eating.

Multilevel regression models were employed to examine associations between baseline frequency of snack occasions and outcome variables at baseline and follow-up. At baseline, greater morning snacking was associated with lower self-reported BMI, higher fruit and vegetable intake, and lower obesogenic dietary index score (Table 4). Greater mid-day snacking was associated with higher fruit and vegetable intake, lower obesogenic dietary index score, and higher levels of distracted-eating at baseline. Greater evening snacking was associated with higher self-reported BMI, higher obesogenic dietary index score, and higher levels of distracted-eating at baseline (Appendix A). At follow-up, higher baseline morning snacking was associated with higher fruit and vegetable intake (Table 5). Higher baseline mid-day snacking was associated with higher fruit and vegetable intake and lower obesogenic dietary index score at follow-up. Higher baseline evening snacking was associated with higher self-reported BMI, higher obesogenic dietary index score, and higher levels of distracted-eating at follow-up. For objective measures, only baseline evening snacking was associated with higher BMI and waist circumference at baseline and follow-up (Table 6).

## 4. Discussion

We present associations between snacking patterns, several obesogenic behaviors, and obesity. The snacking patterns may be considered analogous to extra eating because the analysis adjusted for consuming a meal during the same time-period. Notably, we have found that associations depend on the timing of the snack occasion. Specifically, evening snacking was found to be an obesogenic behavior. Higher levels of evening snacking were associated with higher BMI over repeated self-reported and objective measures at baseline and follow-up. Given the high correlation between self-reported and measured BMI at baseline, consistency in these findings is significant. Furthermore, this relationship was also consistent for repeated measures of waist circumference as well as two measures of obesity-related behaviors, namely the obesogenic dietary index [18] and distracted-eating [22,23]. Interestingly, these patterns of associations were not evidenced for morning or mid-day snack occasions. Higher levels of morning or mid-day snacking were not associated with obesity, yet were associated with higher fruit and vegetable intake. Higher mid-day snacking was also associated with lower obesogenic dietary index scores. To our knowledge, these are the first findings to present associations between timing of snack occasion and measures of obesity as well as related behaviors while adjusting for other meals consumed in the same period.

Our findings are consistent with some [4,5,6,26], but not all [27,28], studies noting associations between increased snack frequency and obesity among adults. Our findings are also consistent with others noting positive relationships between evening snacking and obesity [6,9,26] as well as negative relationships between morning and mid-day snacking and obesity [29]. These findings are supported by studies that have noted higher energy-density of snacks overall among U.S. adults [30] as well as higher energy-density of evening snacks and lower energy-density of morning snacks among French adults [31]. Mechanistically, there is a growing body of evidence noting the importance of circadian timing of food intake to maintain proper metabolic function and body weight regulation [32,33]. We extend previous results by not only adjusting for multiple individual characteristics including demographics and physical activity, but by accounting for other eating occasions during the same period as well. 

Our findings are also consistent with observational studies noting positive associations between distracted-eating and subsequent snack intake [34] as well as experimental studies that either introduce eating mindfulness [35,36] or eating distractions [36]. Few studies have examined snacking in relation to fruit and vegetable intake. Kant and colleagues recently reported that fruit, not vegetable, consumption was higher on days when people snacked compared to days when people did not snack [37]. To our knowledge, there are no other studies that have examined relationships between snacking and fast food or soda intake. Comparisons to studies reporting associations between snacking and diet quality may be most relevant, although this body of evidence is also relatively small and also mixed [38,39,40]. It is possible that mixed findings are due to differences in the composition of snack foods which likely vary across populations defined by age [39,40] as well as culture. 

There are several strengths to this study including: a rigorous statistical approach including multivariable adjustment, use of repeated measures, use of physical measurements for weight, height, and waist circumference, and use of a meal-pattern grid for defining eating occasions. This method [13,14] follows best practices for defining eating occasions [28], specifically, allowing the respondent to distinguish between meals and snacks as well as report the time of eating occasions. This study also includes some limitations. We must acknowledge that we cannot disentangle snack timing from snack composition. It is possible, for example, that it is not really evening snacking that is associated with obesity, but the composition of the snacks that are consumed at that time. We also did not collect information about sleep or other measures of circadian disruption which could have validated the importance of snack timing in these findings. BMI and waist circumference may also not be the most accurate measures of obesity, with different errors as well as implications for different race/ethnic groups. We are also limited by attrition of participants over follow-up as well as a smaller sample size for participants providing physical measurements, each of which not only lowers analytic power but also threatens generalizability to the initial total group of baseline responders. Finally, our respondent group itself may not be generalizable to the U.S. general population given its composition of mostly White working adults.

## 5. Conclusions

Snacking is an important behavior to modify for obesity prevention, although additional dimensions of this behavior need to be considered to produce effective and consistent messaging [41]. Clearly, we note that paying attention to the frequency and timing of snacking is important for obesity prevention. However, we must acknowledge that snack composition is also important given associations with specific dietary behaviors for morning, mid-day, and evening snacks. Reducing evening snacks, and therefore intake of more energy-dense foods, may be another important and simple message for population-level obesity prevention efforts. 

## Figures and Tables

**Table 1 nutrients-11-02320-t001:** Summary of Baseline Demographic and Eating Occasion Variables Over Time Among Promoting Activity and Changes in Eating (PACE) Participants, 2005–2009.

	Baseline				Follow-Up			
	Total		Intensive Assessment Subsample		Total		Intensive Assessment Subsample	
	(N = 34; *n* = 2389)		(N = 34; *n* = 568)		(N = 33; *n* = 1151)		(N = 28; *n* = 268)	
	*n*	*% ^a^*	*n*	*% ^a^*	*n*	*% ^a^*	*n*	*% ^a^*
Age (years) ^b^	43.0	0.7	44.3	0.7	44.3	0.6	45.7	0.9
Gender								
Men	1141	47.4	230	40.4	529	46.0	97	36.2
Women	1248	51.9	338	59.6	622	54.0	171	62.8
Race								
White	1845	77.2	466	82.0	886	77.0	219	81.7
Hispanic/Latino	124	5.2	33	5.8	59	5.1	18	6.7
African American	70	2.9	25	4.4	34	3.0	8	3.0
Asian	236	9.9	26	4.6	122	10.6	16	6.0
Other	80	3.4	15	2.6	36	3.1	6	2.1
Missing	34	1.4	3	0.5	14	1.2	1	0.4
Education								
<HS, HS diploma or GED certificate	368	15.4	58	10.2	192	16.7	31	11.6
Some college or technical college	827	34.6	237	41.7	412	35.8	103	38.4
College graduate	820	34.3	190	33.5	379	32.9	96	35.8
Post-graduate	364	15.2	83	14.6	162	14.1	37	13.8
Missing	10	0.4	0	0.0	6	0.5	1	0.4
Manual occupation								
No	1874	77.9	493	86.8	889	77.2	229	85.5
Yes	508	21.1	74	13.0	251	21.8	37	13.8
Missing	24	1.0	1	0.2	11	1.0	2	0.8
Leisure-time PA (METs per week) ^b^	31.1	29.2	28.7	22.8	29.8	25.5	28.5	22.0
Eating Occasions ^b^								
Main meals	1.7	0.6	1.7	0.6	1.7	0.6	1.7	0.5
Light meals	1.0	0.7	1.0	0.7	1.0	0.7	1.0	0.8
Snacks	1.4	1.1	1.6	1.1	1.5	1.1	1.6	1.1
Drink only	0.8	1.0	1.0	1.1	0.8	1.0	1.0	1.1

N, number of worksites; n, number of participants; HS, high school; GED, general education development; PA, physical activity; MET, metabolic equivalent of task. ^a^ Percentages may not sum to 100 due to rounding error. ^b^ Values are expressed as mean (standard error).

**Table 2 nutrients-11-02320-t002:** Spearman’s Correlations Between Eating Occasions, Dietary Behaviors, and Obesity Outcomes Among Promoting Activity and Changes in Eating (PACE) Intensive Assessment Participants at Baseline, 2005–2007.

	1	2	3	4	5	6	7	8	9	10	11	12
1. Morning snack	1.00											
2. Mid-day snack	0.10	1.00										
3. Evening snack	0.02	−0.02	1.00									
4. Morning meal	−0.18	0.19	0.01	1.00								
5. Mid-day meal	0.07	−0.12	0.07	0.09	1.00							
6. Evening meal	−0.05	0.06	-0.13	−0.03	−0.04	1.00						
7. BMI (self-report)	−0.06	−0.07	0.13	−0.04	−0.03	−0.05	1.00					
8. Fruits and vegetables	0.15	0.19	−0.02	0.24	0.13	0.07	−0.20	1.00				
9. Dietary index	0.04	−0.12	0.15	−0.19	0.04	0.01	0.34	−0.26	1.00			
10. Distracted eating	0.07	0.06	0.16	−0.09	−0.03	−0.10	0.08	−0.11	0.11	1.00		
11. BMI (measured)	−0.04	−0.06	0.13	−0.07	−0.02	−0.05	0.96	−0.19	0.35	0.06	1.00	
12. Waist circumference	−0.06	−0.14	0.10	−0.10	−0.06	−0.01	0.84	−0.20	0.32	0.01	0.86	1.00

^a^ Among participants in intensive assessment subsample without missing data at baseline (*n* = 510).

**Table 3 nutrients-11-02320-t003:** Summary of Outcomes at Baseline and 2-Year Follow-up Among Promoting Activity and Changes in Eating (PACE) Participants, 2005–2009.

	Baseline		Follow-Up		Spearman’s Correlation Coefficient
	*Mean*	*SD*	*Mean*	*SD*	
*Self-Reported Measures ^a^*					
BMI (kg/m^2^)	27.4	6.1	27.5	5.8	0.93
Fruit and vegetables (servings/day)	3.2	2.2	3.5	2.2	0.63
Dietary index (times/week)	2.9	3.0	2.5	2.8	0.76
High distracted-eating (%)	32.8	47.0	29.8	45.8	0.39
*Objective Measures ^b^*					
BMI (kg/m^2^)	29.0	6.7	29.5	6.4	0.93
Waist circumference (cm)	90.3	15.9	87.7	15.1	0.90

^a^ Among participants that provided survey data at baseline and follow-up (*n* = 1151). ^b^ Among participants in intensive assessment subsample at baseline and follow-up (*n* = 268).

**Table 4 nutrients-11-02320-t004:** Associations between baseline snack occasions and self-reported body mass index and related behaviors among PACE adults at baseline (*n* = 2389).

	BMI (kg/m^2^) ^a^ Mean (95% CI)	*p*-Value	Fruit & Veg (Servings/Day) ^a^ Mean (95% CI)		Dietary Index (Times/Week) ^a^ Mean (95% CI)	*p*-Value	High Distracted-Eating (%) Mean (95% CI)	*p*-Value
Morning snack ^b^		0.04		< 0.0001		<0.0001		0.16
0	27.0 (26.7, 27.4)		2.5 (2.4, 2.6)		2.2 (2.1, 2.4)		33.8 (30.9, 36.8)	
1	26.7 (26.3, 27.1)		2.9 (2.8, 3.1)		2.0 (1.8, 2.2)		31.1 (27.7, 34.5)	
2+	26.1 (25.1, 27.2)		3.3 (2.9, 3.7)		1.6 (1.2, 1.9)		30.0 (20.5, 39.5)	
Mid-day snack ^c^		0.68		< 0.0001		0.009		0.02
0	27.0 (26.6, 27.3)		2.5 (2.4, 2.6)		2.2 (2.0, 2.4)		31.1 (28.1, 34.1)	
1	26.7 (26.3, 27.1)		2.8 (2.7, 3.0)		2.0 (1.8, 2.2)		33.2 (29.9, 36.4)	
2+	27.2 (26.2, 28.2)		3.3 (2.9, 3.7)		1.8 (1.5, 2.2)		45.3 (36.8, 53.8)	
Evening snack ^d^		< 0.0001		0.03		<0.0001		<0.0001
0	26.5 (26.1, 26.8)		2.6 (2.5, 2.7)		2.0 (1.8, 2.2)		27.9 (25.1, 30.7)	
1	27.5 (27.0, 27.9)		2.8 (2.7, 3.0)		2.3 (2.1, 2.6)		39.7 (36.3, 43.2)	
2+	27.9 (26.7, 29.1)		2.6 (2.2, 3.0)		2.9 (2.3, 3.5)		44.4 (34.4, 54.3)	

^a^ Log-transformed for analysis; mean values have been back-transformed for presentation. ^b^ Adjusted for age, gender, race/ethnicity, education, manual occupation, leisure-time PA score, and main or light meal reported between 12:00 a.m. and 10:59 a.m. ^c^ Adjusted for age, gender, race/ethnicity, education, manual occupation, leisure-time PA score, and main or light meal reported between 11:00 a.m. and 4:29 p.m. ^d^ Adjusted for age, gender, race/ethnicity, education, manual occupation, leisure-time PA score, and main or light meal reported between 4:30 p.m. and 11:59 p.m.

**Table 5 nutrients-11-02320-t005:** Associations between baseline snack occasions and self-reported body mass index and related behaviors at 2-year follow-up among PACE adults (*n* = 1151).

	BMI (kg/m^2^) ^a^ Mean (95% CI)	*p*-Value	Fruit & Veg (Servings/Day) ^a^ Mean (95% CI)		Dietary Index (Times/Week) ^a^ Mean (95% CI)	*p*-Value	High Distracted-Eating (%) Mean (95% CI)	*p*-Value
Morning snack ^b^		0.79		0.001		0.54		0.72
0	26.7 (26.5, 27.5)		2.8 (2.6, 3.0)		1.7 (1.5, 2.0)		29.4 (25.7, 33.0)	
1	27.1 (26.5, 27.6)		3.2 (3.0, 3.5)		1.7 (1.5, 2.0)		30.5 (26.4, 34.6)	
2+	26.3 (24.9, 27.9)		3.5 (2.9, 4.2)		1.5 (1.1, 2.1)		29.2 (16.2, 42.1)	
Mid-day snack ^c^		0.95		< 0.0001		0.01		0.22
0	27.0 (26.5, 27.5)		2.8 (2.6, 3.0)		1.9 (1.7, 2.1)		28.4 (24.6, 32.1)	
1	26.8 (26.3, 27.3)		3.2 (3.0, 3.5)		1.5 (1.3, 1.8)		31.3 (27.2, 35.5)	
2+	27.9 (26.4, 29.4)		3.7 (3.1, 4.4)		1.7 (1.3, 2.3)		32.8 (20.7, 44.9)	
Evening snack ^d^		< 0.0001		0.17		< 0.0001		0.001
0	26.3 (25.9, 26.8)		2.9 (2.7, 3.1)		1.6 (1.4, 1.8)		25.8 (22.3, 29.3)	
1	28.0 (27.4, 28.6)		3.2 (2.9, 3.4)		2.0 (1.7, 2.2)		35.7 (31.1, 40.2)	
2+	28.0 (26.3, 29.9)		2.9 (2.3, 3.7)		2.5 (1.8, 3.4)		35.8 (21.5, 50.1)	

^a^ Log-transformed for analysis; mean values have been back-transformed for presentation. ^b^ Adjusted for age, gender, race/ethnicity, education, manual occupation, leisure-time PA score, and main or light meal reported between 12:00 a.m. and 10:59 a.m. ^c^ Adjusted for age, gender, race/ethnicity, education, manual occupation, leisure-time PA score, and main or light meal reported between 11:00 a.m. and 4:29 p.m. ^d^ Adjusted for age, gender, race/ethnicity, education, manual occupation, leisure-time PA score, and main or light meal reported between 4:30 p.m. and 11:59 p.m.

**Table 6 nutrients-11-02320-t006:** Associations between baseline snacking occasions and objective measures of body mass index and waist circumference at baseline and 2-year follow-up among PACE adults participating in the intensive assessment subsample.

	Baseline				Follow-Up			
	BMI (kg/m^2^) ^a^ (n = 567) Mean (95% CI)	*p*-Value	Waist Circumference (cm) ^a^ (n = 551) Mean (95% CI)	*p*-Value	BMI (kg/m^2^) ^a^ (n = 262) Mean (95% CI)	*p*-Value	Waist Circumference (cm) ^a^ (n = 258) Mean (95% CI)	*p*-Value
Morning snack ^b^		0.23		0.21		0.43		0.37
0	28.7 (28.0, 29.4)		89.7 (87.9, 91.5)		29.1 (28.2, 30.2)		87.4 (84.9, 90.0)	
1	28.1 (27.3, 28.8)		88.4 (86.5, 90.4)		28.6 (27.5, 29.7)		85.4 (82.8, 88.2)	
2+	28.0 (26.0, 30.2)		87.1 (82.2, 92.4)		28.3 (25.5, 31.4)		86.5 (80.0, 93.5)	
Mid-day snack ^c^		0.48		0.48		0.11		0.68
0	28.4 (27.7, 29.1)		89.6 (87.7, 91.4)		28.4 (27.4, 29.4)		87.4 (84.6, 90.4)	
1	28.4 (27.6, 29.1)		88.3 (86.5, 90.3)		29.1 (28.1, 30.2)		84.9 (82.1, 87.7)	
2+	28.6 (26.7, 30.7)		89.3 (84.6, 94.2)		31.0 (28.1, 34.2)		91.1 (84.0, 98.7)	
Evening snack ^d^		0.007		0.01		0.001		<0.0001
0	27.7 (27.1, 28.4)		87.7 (86.0, 89.4)		28.1 (27.2, 29.0)		84.3 (82.1, 86.6)	
1	29.2 (28.4, 30.0)		90.6 (88.6, 92.6)		29.4 (28.3, 30.6)		88.6 (85.9, 91.4)	
2+	30.1 (27.4, 33.0)		93.3 86.7, 100.3)		34.8 (31.0, 39.1)		99.5 (90.9, 108.9)	

^a^ Log-transformed for analysis; mean values have been back-transformed for presentation. ^b^ Adjusted for age, gender, race/ethnicity, education, manual occupation, leisure-time PA score, and main or light meal reported between 12:00 a.m. and 10:59 am. ^c^ Adjusted for age, gender, race/ethnicity, education, manual occupation, leisure-time PA score, and main or light meal reported between 11:00 a.m. and 4:29 p.m. ^d^ Adjusted for age, gender, race/ethnicity, education, manual occupation, leisure-time PA score, and main or light meal reported between 4:30 p.m. and 11:59 p.m.

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
