# Peer review of "Eating Occasions, Obesity and Related Behaviors in Working Adults: Does it Matter When You Snack?"

_nutrients, 2019, doi:10.3390/nu11102320_

Round 1
Reviewer 1 Report
This manuscript seems just about ready for publication. I'd recommend that a careful reading be done to make sure that English grammar and syntax were followed.
Some comments follow:
The fundamental research question addressed in this manuscript is important and timely, addressing the relationships among snacking behavior and obesity. A clearer articulation of the research question would be helpful to the clarity of the manuscript.
The multilevel statistical methodology is appropriate to the data owing to their nested structure of the group-randomized controlled design, with adequate power from numbers of level-1 and level-2 units.
The finding of poorer weight outcomes with evening snacking and better fruits and vegetables consumption is important for planning interventions to target healthier eating and may help inform policy approaches to obesity control and public health planning. The authors' results provide original information and their unique insights add new perspectives to the research corpus.
Although the text will need some minor grammatical adjustments, it is readable and shows promise to be cited widely. It would be helpful if the authors took note of the limitations of BMI as an obesity metric.
Author Response
Reviewer #1:
This manuscript seems just about ready for publication. I'd recommend that a careful reading be done to make sure that English grammar and syntax were followed.We have reviewed and corrected identified grammatical errors.
The fundamental research question addressed in this manuscript is important and timely, addressing the relationships among snacking behavior and obesity. A clearer articulation of the research question would be helpful to the clarity of the manuscript.We have reworked the last paragraph of the introduction to make the research question and its justification clearer.
It would be helpful if the authors took note of the limitations of BMI as an obesity metric.We have added this as a study limitation.

Reviewer 2 Report
Manuscript ID: Nutrients 587555
Overall comment:
The present manuscript describes the findings of a secondary analysis of data from the PACE study to assess the association between timing of snacking and obesity-related variables. The authors should be commended on an informative an well-written manuscript. There are a few general issues as well as some minor revisions which should be addressed:
The introduction would benefit from a definition of terms like eating occasions in general and snacking specifically. While there is of course colloquial understanding of the difference between a snack and a meal, the authors missed an opportunity to operationally define these terms within the scope of their research. Further, if possible, the authors should expand the available evidence on impact of snacking on overweight and obesity as well as the prevalence of snacking behaviors.
The results clearly demonstrate an association between evening snacking and markers of adiposity. While I understand that the authors adjusted their findings for meal consumption within each time period (morning/mid-day/evening), I am curious whether this model also accounted for overall snacking as well food intake. For example, are people who snack more in the evening also likely to snack more in the morning or at mid-day and/or if their overall food intake (in kcal/d) is also higher? These would be important questions as well, as they may point towards us different messages for obesity prevention (e.g. “if you snack, snack in the morning” vs. “don’t snack at all”). I acknowledge that this may be beyond the scope of this manuscript, but I would appreciate the authors to elaborate on this question.
Given the discrepancies in findings for self-reported and measured BMI as well as differences for self-reported BMI between baseline (Table 3) and follow-up (Table 4), I would recommend eliminating self-reported BMI from the analysis. We know that self-reported BMI is inferior to measured BMI and waist circumference. Since these variables are available, why bother to dilute your findings with an inferior outcome? Especially since the paper did not focus on the discrepancy between the two BMI variables.
Minor comments:
Line 53-54: Move this sentence to the methods Study objective: What were the main hypotheses tested? Line 86: Please expand the explanation of the assessment of eating occasions. Also, please your operational definition of a snack and a meal. Is your definition based on the participants’ definition or did you use specific criteria (e.g. amount of kcal consumed, time spent eating etc.) in the analysis? Line 94: Given the authors’ criticism of the use of pre-defined times (line 40-41), how do they justify the use of three distinct period. Also, consider merging with the first sentence of the statistical analysis Line 98-99: Similar to my comment above, how did the authors define a meal? Specifically, how would they analyze data from a participant who consumes a smaller breakfast (coffee and fruit at 6 am) followed by a bagel at 9 am? What would be the snack, what would be the breakfast? Line 111-112: Move this sentence to introduction if necessary Line 129: Move this sentence to introduction if necessary Line 148-150: More out of curiosity – would it have been possible to express time relative to a person’s individual sleeping pattern rather than using one-size-fits-all approach? Table 1: The header n and % doesn’t make sense for age in years Table 1: What is the difference between a main meal and a light meal? Table 2: What is the correlation between snacking/eating occasions at baseline and follow-up? This data should be included in the manuscript. Line 213: delete the quotation marks. Further consider rephrasing this based on my comments about the definition of snacking. Line 217: consider rephrasing based on whether self-reported BMI is omitted. Line 235-243: While there is nothing with this paragraph per se, it seems somewhat out of place and too detailed, especially when speaking about the experimental studies. On the other hand, is there literature on distracted eating and meal time? For example, are dinners more likely to be eaten while distracted when compared to breakfast or lunch? Lines 272-275: This speculation came out of nowhere – how did the authors make the connection to 5+ fruit/vegetable intake? While more snacking in the morning is associated with more fruit & vegetable consumption, I do not see any evidence in this manuscript that supports the notion that greater fruit and vegetable consumption is linked to less snacking and specifically evening snacking (unless I missed something completely). I strongly consider revising the conclusions so they line up with the remaining manuscript.
Side comment: Although the presentation of the data is overall well, the manuscript might benefit from a figure that graphically displays the major finding(s).
Author Response
Reviewer #2:
The introduction would benefit from a definition of terms like eating occasions in general and snacking specifically. While there is of course colloquial understanding of the difference between a snack and a meal, the authors missed an opportunity to operationally define these terms within the scope of their research. Further, if possible, the authors should expand the available evidence on impact of snacking on overweight and obesity as well as the prevalence of snacking behaviors.Taking these comments in reverse order, let us first address the question of defining “snacking”. H. Berteus Forslund and colleagues1 originally developed a grid assessment method to allow respondents to check the type of eating occasion and write in the time when it was consumed. We have tried to explain this more clearly in the methods section. The definition of a snack is therefore whatever the respondent thinks it is. The layout of the matrix or grid makes this rather clear. We are proposing including the question, example and grid that we used in our surveys, adapting the Berteus Forslund method, as Supplementary Material with this paper.
Secondly, we have reworked the first and second paragraphs of the introduction to explicitly define eating occasions. We built on our original discussion of how studies have been defining meals, including snacks. We discussed the limitations of these approaches as well as how the grid assessment method of eating occasions used in this study can address those limitations, specifically by having the respondent assign meal categories and time to their own intakes. We have done an extensive literature review of the impact of snacking on overweight and obesity as well as dietary behaviors among adults. As a result, we added an additional citation by Mesas and colleagues (2012), but note that this body of literature, in relation to adult behaviors, is rather small.
The results clearly demonstrate an association between evening snacking and markers of adiposity. While I understand that the authors adjusted their findings for meal consumption within each time period (morning/mid-day/evening), I am curious whether this model also accounted for overall snacking as well food intake. For example, are people who snack more in the evening also likely to snack more in the morning or at mid-day and/or if their overall food intake (in kcal/d) is also higher? These would be important questions as well, as they may point towards us different messages for obesity prevention (e.g. “if you snack, snack in the morning” vs. “don’t snack at all”). I acknowledge that this may be beyond the scope of this manuscript, but I would appreciate the authors to elaborate on this question.The reviewer is correct to infer that our analyses did not attempt an adjustment for overall intake. Specifically, the evaluation of caloric intake is beyond the scope of this manuscript : we are intentionally moving away from nutrient-based approaches given the complexities and error associated with them. Nonetheless, using the reviewer’s example, one way to determine whether evening snackers are also likely to be morning snackers, is to look at correlations between snacking variables. We have created a new table (new Table 2) with this information, and additional correlations between outcome variables at baseline. Here, we see through the small correlation coefficient, that people who snack in the morning are not the same people who snack in the evening, in general.
We do acknowledge that it is difficult to disentangle whether it is really snacking at certain times of day that is associated with obesity or whether it is the type of foods that tend to be consumed at different times of day. We think our analyses examining associations of snacking during morning, mid-day, and evening with obesity-related dietary behaviors helps to answer this question. Specifically, the dietary behaviors associated with snacking at different times of day are very different: those who eat more morning snacks also eat more servings of fruits and vegetables whereas those who snack in the evening tend to eat more fast food meals including French fries and soft-drinks. This result would tend to support the message “If you snack, snack on fruits and vegetables”.
Given the discrepancies in findings for self-reported and measured BMI as well as differences for self-reported BMI between baseline (Table 3) and follow-up (Table 4), I would recommend eliminating self-reported BMI from the analysis. We know that self-reported BMI is inferior to measured BMI and waist circumference. Since these variables are available, why bother to dilute your findings with an inferior outcome? Especially since the paper did not focus on the discrepancy between the two BMI variables.Findings for associations between evening snacking and BMI are consistent between baseline and follow-up for both self-reported and objective measures. Discrepancies in findings for self-reported and objective measures could be either attributable to measurement error or to differences between the subsample and the whole population of baseline responders. We have added a table of correlations between snacking and outcome measures to support that measures of BMI are highly correlated. It seems prudent to include both results.
Line 53-54: Move this sentence to the methods
Done.
Study objective: What were the main hypotheses tested?We have reworked the last paragraph of the introduction to make the research question and relationships examined more clear. We did not explicitly state a priori hypotheses given mixed findings in the current body of literature.
Line 86: Please expand the explanation of the assessment of eating occasions. Also, please your operational definition of a snack and a meal. Is your definition based on the participants’ definition or did you use specific criteria (e.g. amount of kcal consumed, time spent eating etc.) in the analysis?Again, the eating pattern grid method allows the respondent to categorize their own intake as a main or light meal, snack, or drink. The grid provides example foods that fit into each of these categories, but it is ultimately up to the respondent to make that determination. Our operational definition of a meal is all eating occasions marked as a meal by the respondent. All snacks are defined as all eating occasions marked as a snack by the respondent.
Line 94: Given the authors’ criticism of the use of pre-defined times (line 40-41), how do they justify the use of three distinct period. Also, consider merging with the first sentence of the statistical analysisThe generation of time periods was empirical and not pre-defined by the authors. We have moved the description of snack periods to that variable description and out of the statistical analysis section.
Line 98-99: Similar to my comment above, how did the authors define a meal? Specifically, how would they analyze data from a participant who consumes a smaller breakfast (coffee and fruit at 6 am) followed by a bagel at 9 am? What would be the snack, what would be the breakfast?Please see our response to reviewer comment #6. The light meal and snack would be defined by the respondent and they could submit as many entries as they wanted. It is possible that respondents could report >1 meal and >1 snack per period. Table 1 provides mean values of each type of eating occasion.
Line 111-112: Move this sentence to introduction if necessary.We will leave the sentence in the Methods section as it really pertains to the variable description rather than the narrative setting up the analysis.
Line 129: Move this sentence to introduction if necessary.We have taken the reviewer’s suggestion to move the first sentence of this variable description to the introduction, but have decided to keep the second sentence per our response to reviewer’s comment #9.
Line 148-150: More out of curiosity – would it have been possible to express time relative to a person’s individual sleeping pattern rather than using one-size-fits-all approach?No. We did not measure sleep in this study.
Table 1: The header n and % doesn’t make sense for age in years.There is a footnote in Table 1 to alert the reader to that difference which is standard practice.
Table 1: What is the difference between a main meal and a light meal?Types of eating occasions (including main and light meals) are described in the Methods under the “Eating Occasions” section.
Table 2: What is the correlation between snacking/eating occasions at baseline and follow-up? This data should be included in the manuscript. Line 213: delete the quotation marks. Further consider rephrasing this based on my comments about the definition of snacking.We have deleted the quotations marks. We still maintain that our analyses still examine the impact of extra snacking—within time periods rather than over the entire day.
Line 217: consider rephrasing based on whether self-reported BMI is omitted.We are keeping self-reported BMI in these analyses as we feel strongly that it is a more robust variable given its greater number of observations and is also very highly correlated with our objective measure among a smaller group of participants. We have added this point to the first paragraph of the Discussion.
Line 235-243: While there is nothing with this paragraph per se, it seems somewhat out of place and too detailed, especially when speaking about the experimental studies.We have deleted the detailed description of the experimental studies.
On the other hand, is there literature on distracted eating and meal time? For example, are dinners more likely to be eaten while distracted when compared to breakfast or lunch?The experiments studies described did indicate that distracted eating influenced subsequent meal intake, but did not report whether this differed by meal type. However, we have removed this discussion from the manuscript per reviewer comment #17.
Lines 272-275: This speculation came out of nowhere – how did the authors make the connection to 5+ fruit/vegetable intake? While more snacking in the morning is associated with more fruit & vegetable consumption, I do not see any evidence in this manuscript that supports the notion that greater fruit and vegetable consumption is linked to less snacking and specifically evening snacking (unless I missed something completely). I strongly consider revising the conclusions so they line up with the remaining manuscript.Side comment: Although the presentation of the data is overall well, the manuscript might benefit from a figure that graphically displays the major finding(s).
We thank the reviewer for this suggestion.
Berteus Forslund H, Lindroos AK, Sjostrom L, Lissner L. Meal patterns and obesity in Swedish women-a simple instrument describing usual meal types, frequency and temporal distribution. European journal of clinical nutrition. 2002;56(8):740-747.